# Intravenous MSC-Treatment Improves Impaired Brain Functions in the R6/2 Mouse Model of Huntington’s Disease via Recovered Hepatic Pathological Changes

**DOI:** 10.3390/cells13060469

**Published:** 2024-03-07

**Authors:** Libo Yu-Taeger, Ali El-Ayoubi, Pengfei Qi, Lusine Danielyan, Hoa Huu Phuc Nguyen

**Affiliations:** 1Department of Human Genetics, Ruhr University of Bochum, D-44801 Bochum, Germany; 2Institute of Medical Genetics and Applied Genomics, University of Tuebingen, D-72076 Tuebingen, Germany; 3Department of Clinical Pharmacology, University Hospital of Tuebingen, D-72076 Tuebingen, Germany; 4Departments of Biochemistry and Clinical Pharmacology, and Neuroscience Laboratory, Yerevan State Medical University, Yerevan 0025, Armenia; 5Department of Medical Chemistry, Yerevan State Medical University, Yerevan 0025, Armenia

**Keywords:** Huntington disease, mesenchymal stromal cells, intravenous, R6/2 mice, liver function, hepatic pathology, neuropathology, inflammation

## Abstract

Huntington’s disease (HD), a congenital neurodegenerative disorder, extends its pathological damages beyond the nervous system. The systematic manifestation of HD has been extensively described in numerous studies, including dysfunction in peripheral organs and peripheral inflammation. Gut dysbiosis and the gut–liver–brain axis have garnered greater emphasis in neurodegenerative research, and increased plasma levels of pro-inflammatory cytokines have been identified in HD patients and various in vivo models, correlating with disease progression. In the present study, we investigated hepatic pathological markers in the liver of R6/2 mice which convey exon 1 of the human mutant huntingtin gene. Furthermore, we evaluated the impact of intravenously administered Mesenchymal Stromal Cells (MSCs) on the liver enzymes, changes in hepatic inflammatory markers, as well as brain pathology and behavioral deficits in R6/2 mice. Our results revealed altered enzyme expression and increased levels of inflammatory mediators in the liver of R6/2 mice, which were significantly attenuated in the MSC-treated R6/2 mice. Remarkably, neuronal pathology and altered motor activities in the MSC-treated R6/2 mice were significantly ameliorated, despite the absence of MSCs in the postmortem brain. Our data highlight the importance of hepatic pathological changes in HD, providing a potential therapeutic approach. Moreover, the data open new perspectives for the search in blood biomarkers correlating with liver pathology in HD.

## 1. Introduction

Huntington disease (HD), a congenital progressive neurodegenerative disease, is attributed to an expanded CAG repeat in the N-terminal region of the Huntingtin gene (*HTT*). Mutant HTT primarily affects the striatum and leads to abnormal aggregates, neuronal dysfunction, and ultimately death. Clinical manifestations of HD include motor dysfunction, cognitive impairment, psychiatric disturbance, and frequently observed sleeping disorders [1,2,3,4]. Beyond neurological symptoms, numerous studies describe peripheral symptoms and systemic manifestations of HD, encompassing metabolic dysregulation, peripheral inflammation, and dysfunction in various organs. Pathological events in the peripheral organs have been observed in the heart, muscle, bone, adrenal glands, gut, and liver [5,6].

The liver, a crucial organ with diverse functions, is implicated in the abnormalities associated with HD symptoms [7,8]. The liver represents the most important metabolic organ, controlling various metabolic processes such as glucose metabolism and glycogen storage. Furthermore, as a blood-filtering organ, the liver plays an essential role in immune surveillance, removing pathogens, cellular debris, and foreign particles to maintain overall immune homeostasis. In addition, the liver can influence the sleep–wake cycle by synchronizing its circadian rhythm with the master circadian clock in the brain [9]. Moreover, disruptions in these rhythms can impact overall circadian regulation, potentially influencing sleep. Consistently across various clinical studies, elevated lactate concentrations in the blood have been reported in patients with HD [10,11,12], suggesting impaired lactate utilization by the liver. Furthermore, increased levels of liver Gamma-Glutamyl-Transferase (GGT) in symptomatic HD patients further indicate compromised liver function [8]. These observations align with findings in R6/2 HD mice, where reduced lactate clearance, gluconeogenesis, and urea cycle deficiency have been documented [13,14]. As observed in the brain of pre-manifest and manifest HD patients, progressive mitochondrial dysfunction is also evident in the hepatic cells of these patients [7,15]. 

Recent studies propose that the liver impacts the risk and progression of neurodegenerative disorders [16,17,18,19]. As the peripheral organs and the central nervous system (CNS) bidirectionally communicate, changes in one can influence the other. Understanding this interaction may provide insight into potential therapeutic targets for HD. In our study, we thoroughly examine the pathological features in the liver of R6/2 HD mice and specifically address the impact of liver-targeting treatment on the CNS. 

As a therapeutic option for liver dysfunction, Mesenchymal Stromal Cells (MSCs) have been increasingly recognized due to their ability to promote liver regeneration and inhibit the progression of liver fibrosis [20]. Furthermore, MSCs are continuously investigated clinically and preclinically for neurodegenerative disorders [21,22], leveraging some of their therapeutic properties such as secreting glia cell-derived neurotrophic factors (GDNFs), brain-derived neurotrophic factor (BDNF), as well as vascular endothelial growth factor (vEGF) [23,24]. Furthermore, the distinctive immunomodulatory properties of MSCs exert beneficial effects in the inflammatory pathology commonly observed in neurodegenerative diseases [25,26]. The mechanisms of MSC-mediated immunomodulation involve direct contact or the release of soluble immune regulators through the MSC secretome, thereby modulating immune responses and facilitating immunosuppression [27]. R6/2 HD transgenic mice express exon1 of human mutant Huntingtin with approximately 140 CAG repeats [28]. These mice exhibit early-onset, robust progressive neuropathological changes, behavioral phenotypes [29], and a wide range of dysfunction in peripheral organs and tissues [14,30,31,32,33,34].

To investigate the liver pathology and its impact on the CNS in HD, we comprehensively assessed enzyme levels and inflammatory markers in the liver along with neuronal markers in the brain and behaviors related to the brain function. Furthermore, we administered MSCs intravenously to R6/2 mice, as it is the optimal route for MSC migration to the liver and aids in liver repair [35]. All hepatic and neuropathological changes were compared between treated and non-treated animals. Our findings underscore the significant impact of liver pathology on the CNS in HD.

## 2. Materials and Methods

### 2.1. Animals and Genotyping

All experiments were performed on female mice. Mice of different genotypes were housed under 12 h light and dark cycles with accessible water and food. The Regierungspraesidium Tuebingen local ethics committee had approved all animal experiments (License Number: HG 7/16), which were conducted in agreement with the German Animal Welfare Act and followed the guidelines of the Federation of European Laboratory Animal Science Associations (Directive 2010/63/EU). 

Wild-type B6CBAF1/J males were crossed with ovary-transplanted R6/2 females (B6CBA-TgN (Hdexon1)62Gbp/J) for breeding purposes (Charles River Laboratory). Then, female R6/2 and wild-type (WT) mice were divided into three different treatment groups. Group (1): MSC-treated R6/2 mice (R6/2-MSC); Group (2): Phosphate-buffered saline (PBS)-treated R6/2 mice (R6/2-PBS) and; Group (3): PBS-treated WT mice (WT-PBS). Animals were separated into different groups according to their body weight and rotarod test performance at 3 weeks of age in order to counterbalance the potential litter effects. Then, 6.5 weeks post-MSC vs. PBS treatment, animals were sacrificed for postmortem analyses. The mice were sacrificed five days post-MSC delivery for analyzing cell migration in their brains.

### 2.2. Intravenous Injection

To maximize therapeutic efficacy, repeated intravenous injections were performed on 4-week-old and 6.5-week-old mice via the tail vein using a 30 G needle. At the respective ages, R6/2-MSC mice received in total 2 million MSCs divided in two injections (for each injection, 1 million MSCs were resuspended in 50 µL PBS) with a 2-day interval between each administration. Mice of both PBS groups only received the same volume of PBS. Bone marrow-derived MSCs were harvested from the tibia and femur [36] of eGFP-expressing mice (C57BL/6-Tg(UBC-GFP)30Scha/J mice, Jackson Laboratories, Bar Harbor, ME, USA) and cultured [37] as described previously.

### 2.3. Rotarod Test

Motor function was investigated using rotarod tests as previously described [37]. Examination of the mice was conducted at 3, 6 and 10 weeks of age (1 week prior to MSC-administration for dividing mice into groups, 2 and 6 weeks after the first administration of MSCs for testing the therapeutic potential.

### 2.4. LabMaster

We utilized the LabMaster system (TSE system GmbH, Berlin, Germany) to monitor the locomotor activities of the mice. This system provides a home cage-like environment fixed in an infrared light fame. Fine movement was defined by repeated breaks of the same light beam along the x and y axes, while breaks of alternate barriers along the x and y axes were defined as ambulatory movements. Animals were monitored at 5 and 9 weeks of age, each time point, for 22 h. Data were analyzed by adding all activities either in the light phase or dark phase individually. 

### 2.5. RNA Extraction and qPCR

We conducted RNA extraction, cDNA synthesis and quantification PCR as described previously [38]. Mice at 10.5 weeks (6.5 weeks after the first MSC administration) were sacrificed by CO_2_ asphyxiation; then, the brains and livers were rapidly harvested. Brain regions and the right medial lobe of the liver were dissected on ice; then, tissues were stored at −80 °C after snap-freezing in liquid nitrogen. The total RNA was extracted using the Rneasy Kit (#217004, Qiagen, Hilden, Germany) following the manufacturer’s protocols. Afterwards, 1 µg of total RNA was used for the reverse transcription reaction (QuantiTect Reverse Transcription kit, #218061, Qiagen, Germany). We used the Pfaffl model to calculate relative expression [39] after normalization to the geometric mean relative expression of three reference genes (Sdha, Gapdh and Hprt1 for liver and spleen).

### 2.6. Immunofluorescence Staining

Immunofluorescence stainings of mouse liver using fresh tissue were performed as described in a comparable study on the treatment effects of MSCs in liver pathology [40]. First, 16 µm thick sections were cut at −20 °C using a cryostat and directly mounted on the microscope slides. After the blocking step with 5% normal serum in TBS with 0.3% Triton X-100, sections were incubated in primary antibody at room temperature overnight with anti-Huntingtin (EM48, MAB5374, Millipore, Darmstadt, Germany) diluted 1:500, anti-GFP (NB600-308S, Novus Biotech, Centennial, CO, USA) diluted 1:300, anti-DARPP-32 (ab40801, Abcam, Cambridge, UK) diluted 1:5000, and anti-TH (#657012, Merck, Darmstadt, Germany) diluted 1:1000. Incubation of the secondary antibodies Goat Anti-Rabbit IgG H&L (Alexa Fluor^®^ 594, ab150080, Abcam, Germany) and Donkey Anti-Mouse IgG H&L (Alexa Fluor^®^ 594) (#715-585-150, Dianova, Hamburg, Germany) both diluted 1:400 were carried out at RT for 1.5 h. 

To enhance data comparability, immunofluorescence stainings of mouse brain sections were performed using the protocol outlined in our previous publication [37], which specifically addressed the impact of MSC treatment on brain pathology. Mouse brains were fixed via transcardiac perfusion with 4% PFA in PBS. We then serially cryo-sectioned 25 µm coronal sections at 150 µm intervals with every 6th striatal brain section (from Bregma 1.54 to 0.14 mm [41]) chosen for free-floating fluorescence staining to analyze each of the following target proteins: GFP, mHTT inclusion bodies, dopamine- and cAMP-regulated neuronal phosphoprotein (DARPP-32) and tyrosine hydroxylase (TH) individually. In brief, following the blocking step in TBS containing 5% normal serum, brain sections were incubated in primary antibodies anti-GFP (NB600-308S, Novus Biotech, USA, 1:500), anti-Huntingtin (EM48, MAB5374, Millipore, Germany 1:1000), anti-DARPP-32 (ab40801, Abcam, Germany, 1:2000) and anti-TH (#657012, Merck, Germany, 1:1000) at room temperature overnight. The same secondary antibodies and conditions as for staining liver sections were applied. All images were acquired and analyzed using the Zeiss Axioplan2 fluorescence microscope and Java 6 Fiji (National Institutes of Health, Bethesda, MD, USA), respectively. 

### 2.7. Protein Extraction and Western Blot Analyses

Frozen fresh tissue was homogenized in RIPA buffer supplemented with proteinase inhibitor cocktail complete without EDTA (#11836170, Roche Diagnostics, Penzberg, Germany). Western blot analysis was performed as previously described [38]. Primary antibodies were used at a dilution of 1:1000 for DARPP-32 (ab40801, Abcam, Germany), 1:1000 for TH (#657012, Merck, Germany) and 1:500 for F4/80 (#70076, cell signaling, Germany). Fluorescent-conjugated secondary antibodies were applied, and fluorescence was detected and then quantified using an ODYSSEY FC imaging system with Image Studio software (Version 5.2) (LI-COR Biosciences, Lincoln, NE, USA). 

### 2.8. Statistical Analyses

Priori Power Analysis (Statistical Solutions, LLC, Beavercreek, OH, USA; Power & Sample Size Calculator) was used to determine the sample size. All statistical analyses were performed using GraphPad Prism 9.1.2. Data are presented as the individual values and mean ± SEM for each condition. Repeated ANOVA measures followed by Tukey’s multiple comparison tests were used for the longitudinal analysis of rotarod performance and activities screened by LabMaster. For the remaining comparisons, two-tailed unpaired Student’s *t*-tests were used for the analyses of protein and RNA expression levels. *p* < 0.05 was considered statistically significant.

## 3. Results

### 3.1. Pathological and Functional Changes in R6/2 Mice Liver

Aggregates containing mHTT in the brain is a pathological hallmark of HD: Neuropil aggregates as well as ample nuclear inclusion bodies have also been found in the striatum and cortex of R6/2 mice [42]. Firstly, we analyzed mHTT aggregate formation in the liver of R6/2 mice at 10.5 weeks of age by the immunofluorescence staining of human huntingtin. A moderate number of aggregates were found in the liver, and these aggregates were slightly smaller in size compared to the nuclear inclusion bodies in the striatum (Figure 1A). These aggregates were completely absent in WT mice.

Alanine aminotransferase (ALT) and aspartate aminotransferase (AST) enzymes are mainly found in the liver. ALT facilitates the conversion of keto acids, whereas AST catalyzes the reversible transfer of an amino group from glutamate to oxaloacetate, thereby contributing to the provision of energy supply. Altered levels of ALT and AST indicate liver damage or an underlying disorder affecting its function. Therefore, we analyzed the mRNA expression levels of *Alt* and *Ast*. Significant reductions were found in the mRNA levels of *Alt* (*p* = 0.0019) and *Ast* (*p* = 0.0144) in R6/2 mice. Furthermore, the protein expression level of ALT was also significantly reduced (*p* = 0.0415) (Figure 1B). Cytochrome P450s (CYPs) are a big family of proteins predominantly expressed in the liver and are crucial for detoxification besides the metabolic activation of xenobiotics [43]. Several studies have reported that mHTT causes dysregulated transcription of *CYPs* in HD patients and animals [44,45,46,47]. Furthermore, a significant body of research indicates that the reduced activity of CYP enzymes is associated with the severity of chronic liver disease with the exception of an irreversible correlation observed in a few of these enzymes [48,49,50]. Hence, we analyzed the mRNA levels of several Cyps in the liver including *Cyp1a2*, *Cyp2a4*, *Cyp2b9*, *Cyp2c29*, *Cyp3a11*, and *Cyp3a41*. Three of six analyzed genes (*Cyp1a*, *Cyp3a11* and *Cyp3a41*) had significantly decreased or increased expression levels in R6/2 mice compared to WT littermates, whereas both *Cyp2b9* (*p* = 0.0709) and *Cyp2a4* (*p* = 0.0975) genes showed a trend toward significance between R6/2 mice and WT controls (Figure 1C). Taken together, our results indicate an altered liver function in R6/2 mice at 10.5 weeks of age.

### 3.2. Elevated Inflammatory Markers in the Liver of R6/2 Mice

The increase in pro-inflammatory cytokines has been observed in the brain and periphery of HD patients and R6/2 mice [51]. To investigate whether the hepatic inflammatory response is elevated in R6/2 mice, we quantified mRNA levels of the pro-inflammatory factors *IL-1ß*, *IL-12*, *Mcp-1*, *Tnf* and *Tgf-β1*, chemokine receptors *Ccr2* and *Ccr5*, chemokine ligands *Cxcl12* and *Cxcl13*, and prostaglandin E2 receptor (*Ptger2*). Statistical analysis showed that mRNA levels of *Tnf*, *Tgf-β1*, *IL-12a* and *Ccr2* were significantly increased in the liver of R6/2 mice compared to WT littermates, while a trend toward significance was observed in *Ccr5* (*p* = 0.0969) (Figure 2). 

### 3.3. MSC Treatment Recovers Hepatic Enzyme Levels and Reduces Hepatic Inflammation in R6/2 Mice

Numerous studies have reported on the success of the MSC therapeutic effect in inflammatory regulation [52] as well as the amelioration of neuropathology in HD [53,54,55]. We applied bone marrow-derived mouse MSCs for the treatment of R6/2 mice via intravenous administration, which has demonstrated to be the best delivery route of MSCs for improving liver function [35]. At 4 weeks and 6.5 weeks of age, the R6/2 mice of the treatment group (R6/2-MSC mice) received repeated administrations: in total 4 million GFP-tagged MSCs. R6/2 mice, which only received PBS (R6/2-PBS mice), served as controls. To assess the therapeutic potential of MSCs on liver dysfunction and the alteration of the hepatic immune response in R6/2 mice, postmortem analyses were performed in 10.5-week-old R6/2 and WT mice. Firstly, we analyzed MSC homing to the brain and liver; 4 weeks after the second administration, there was no GFP signal in the brain of treated mice, while the liver showed an abundantly GFP-positive signal, suggesting the presence of debris or degrading MSCs (Figure 3A). Additionally, we analyzed the hepatic parameters in R6/2-MSC and R6/2-PBS mice. The mRNA levels of the two important enzymes influencing liver function, ALT and AST, did not change between the aforementioned groups. The quantification of the ALT protein expression was consistent with the mRNA analyses (Figure 3A). Two members of the *Cyps*, *Cyp1a2* and *Cyp3a41*, which showed a significant decrease and increase in R6/2 mice, respectively, also showed a slight change in R6/2-MSC mice compared to R6/2-PBS mice with respective p values of 0.0837 and 0.0662. The mRNA levels of *Cyp2a4*, *Cyp2b9* and *Cyp3a11* did not show any changes in the treated mice (Figure 3B). Moreover, we compared the significantly and slightly increased inflammation regulatory factors we found in R6/2 mice vs. WT mice. R6/2-MSC mice exhibited significantly decreased mRNA expression of the *Tgf-β1* gene (*p* = 0.0260) and *IL-12a* (*p* = 0.0086) compared to R6/2-PBS mice. Although there was no statistical significance, a trend toward reduction in mRNA levels in *Tnf* (*p* = 0.0732), *Ccr2* (*p* = 0.0860) and *Ccr5* (*p* = 0.0649) was observed in R6/2-MSC vs. R6/2-PBS mice (Figure 3C). These results demonstrate a significant effect of intravenous administration of MSC therapy on hepatic inflammation regulatory factors in R6/2 mice as well as the gene expression levels of some of the hepatic enzyme Cyps. 

As macrophages play an essential role in promoting local inflammation, it was important to analyze if and to what extent MSC therapy can modulate the proliferation and activation of these cells in the liver of R6/2 mice. Hence, we quantified the protein (F4/80) and mRNA (*Adgre1*) levels of the macrophage markers in R6/2-PBS mice vs. WT-PBS groups and between the two R6/2 treatment groups. F4/80 protein levels were significantly increased in the R6/2-PBS group compared to the WT-PBS group (*p* = 0.0452); this elevation was reduced in the R6/2-MSC group compared to the R6/2-PBS group (*p* = 0.0303). Quantification of mRNA levels showed that the expression level of the *Adgre1* gene (encoding protein F4/80) was slightly increased in R6/2-PBS mice vs. WT-PBS mice (*p* = 0.0591), while the reduction in R6/2-MSC mice vs. R6/2-PBS mice was highly significant (*p* < 0.0001) (Figure 4A,B). Furthermore, we analyzed mRNA levels of *Cd206*, *Cd68* and *Cd163*, markers of the M2-type (anti-inflammation) macrophage, where MSCs promote polarization [52] and in turn increase the population. In all these markers, we did not find any differences in either the R6/2-PBS mice vs. WT-PBS comparisons or between the two R6/2 treatment groups (Figure 4C). 

### 3.4. MSC Treatment Restores the Levels of Neuronal Markers and Microglia Markers in the Striatum of R6/2 Mice

The results above indicate that enhanced inflammation response and diminished liver function were ameliorated after treatment with MSCs via intravenous delivery. Since pro-inflammatory cytokines could enter the CNS through the leaky blood–brain barrier (BBB) in R6/2 mice [56], it raises the question of whether the functional amelioration and abated inflammation in the liver may affect brain pathology. We therefore analyzed the neuronal marker of striatal cells DARPP-32, a dopamine-regulated phosphoprotein, which is expressed in GABAergic spiny projection neurons in the striatum. We also analyzed dopamine neuron projection in the striatum using the dopamine neuronal marker tyrosine hydroxylase (TH). In line with previous studies [37], western blot analyses demonstrated a significant reduction in both DARPP-32 (*p* = 0.0004) and TH (*p* < 0.0001) in R6/2-PBS mice compared to WT-PBS. Comparing R6/2-MSC mice with R6/2-PBS mice, both DARPP-32 (*p* = 0.0395) and TH (*p* = 0.0224) were significantly increased (Figure 5A). The immunofluorescence staining of coronal brain sections (Figure 5B) further confirmed these results. Furthermore, the microglia marker Iba-1 and astrocytes marker GFAP were analyzed in the striatum as well showing significantly decreased protein expression levels of Iba-1 in R6/2-PBS vs. WT-PBS mice (*p* = 0.0009) and significantly increased levels in R6/2-MSC vs. R6/2-PBS mice (*p* = 0.0056), while no change in GFAP expression levels between any groups could be observed (Figure 5A). Collectively, increasing levels of the reduced neuronal markers DARPP-32 and TH in the striatum of R6/2 mice, as well as changes in microglial activation, suggest an ameliorated neuronal function in the MSC-treated group.

### 3.5. Altered Locomotor Activities in R6/2 Mice Were Improved after MSC Treatment

In order to evaluate neuronal function in R6/2 mice and MSC therapeutic potential, we tested motor function using the rotarod test and ambulatory activities using LabMaster at different ages. Rotarod tests were performed at 6 and 10 weeks of age in the 3 experimental groups, WT-PBS, R6/2-PBS and R6/2-MSC. R6/2 mice showed a highly significant decrease in latency to fall during the whole investigation period compared to the WT group (two-way ANOVA, group effect: F(2.39) = 23.59, *p* < 0.0001). Tukey’s post hoc test demonstrated a significant reduction in latency in R6/2-PBS mice vs. WT-PBS mice at both 6 and 10 weeks of age, while no significant difference was detected between MSC-treated and non-treated R6/2 mice (Figure 6A). PhenoMaster was performed with mice at 5 and 9 weeks of ages to track the locomotor activities of the animals for 22 h at each time point. Statistical analysis using two-way ANOVA indicated increased total locomotor activities in the 22 h tracking period for the R6/2 mice (group effect: F(2.39) = 8.034, *p* = 0.0010). Tukey’s post hoc test comparing the average of each groups demonstrated that the differences were further evidenced in R6/2-PBS mice vs. WT-PBS mice (*p* = 0.0009) as well as in R6/2-MSC mice vs. R6/2-PBS mice (*p* = 0.0249) (Figure 6B). Differences in total activity, ambulatory activity and fine movement were analyzed in the light (sleeping phase) and dark phase (active phase) separately. Highly significant group × age interactions were found in the light phase in both ambulatory activities (F(2.35) = 13.92, *p* < 0.0001) and fine movement (F(2.35) = 15.42, *p* < 0.0001). A less significant group x age interaction was also observed in fine movement in the dark phase (F(2.35) = 6.404, *p* = 0.0043). Statistically significant differences in group effect were determined in both parameters for both the light and dark phases, and these differences were attributed to R6/2-PBS vs. WT-PBS mice and both R6/2-treated and non-treated mice (Tukey’s post hoc test) (Figure 6C–F). Overall, locomotor activities were more prominently elevated in the light phase. We conclude that motor function and locomotor activities were altered in the later stage of the R6/2 mice, particularly in the sleeping period. MSC-treatment via intravenous administration was able to ameliorate locomotor activities that were maintained to a comparable level of WT littermates.

## 4. Discussion

In recent years, emerging studies suggest that mechanisms underlying many neurodegenerative diseases extend beyond the brain [57,58,59,60,61]. This includes phenomena such as protein aggregation and mitochondrial dysfunction in peripheral organs, alterations in peripheral metabolism, and peripheral inflammation. The gut–liver–brain axis has garnered significant attention in the field of neurodegenerative diseases [62,63]. Consistently, gut dysbiosis and increased intestinal permeability have also been found in HD mice [30,64,65]. Interestingly, a fecal microbiota transplant is able to ameliorate gut dysbiosis, particularly improving cognitive deficits in HD mice [64]. As a second barrier, the liver filters gut absorbents before they enter systemic circulation. Increased intestinal permeability leads to elevations in the portal influx of bacteria and their products (also called pathogen-associated molecular patterns or PAMPs) further promoting systemic inflammation [66,67,68]. Numerous studies have reported on altered peripheral inflammation in HD patients. Elevated plasma levels of pro-inflammatory markers, including IL-4, IL-5, IL-6, IL-8, IL-10, and TNF-α, have been observed in both HD patients [69] and HD animal models [51,70,71,72]. Additionally, increased plasma levels of several chemokines, such as eotaxin, eotaxin-3, macrophage inflammatory protein-1 β (MIP-1 β), monocyte chemotactic protein-1 (MCP-1), and MCP-4, have been noted. Interestingly, the circulating levels of certain pro-inflammatory cytokines were found to be positively and negatively correlated with motor scores and function scores in HD patients, respectively [69,73]. Some pro-inflammatory markers were found to be elevated in premanifest HD patients and correlated with microglial cell activation [73]. A particular study reported that increased IL-6 was detected approximately 16 years on average before the onset of HD [69].

In alignment, studies indicate gut dysbiosis, gut barrier leakage, and systemic inflammation in R6/2 mice [30,51]. An excessive influx of escaped microbiota and PAMPs may overwhelm the liver’s filtration capacity, leading to lasting damage and triggering a systemic inflammatory response initiated in the liver [74]. Intrinsically, PAMPs activate hepatic-resident macrophage Kupffer cells, resulting in an increased expression of cytokines and chemokines. Indeed, in this study, we observed a significant increase in the Kupffer cell marker F4/80 in R6/2 mice at protein levels, indicating the proliferation and activation of Kupffer cells (Figure 4). Further evidence of elevated levels of cytokines, chemokine receptors, and chemokine ligands in the liver (Figure 2) confirm the heightened inflammation in the liver of R6/2 mice. 

Our results not only unveiled increased inflammation but also altered expression levels of several liver enzymes, including ALT, AST, and various members of the CYP family. ALT and AST are crucial hepatic enzymes involved in energy homeostasis [75,76] and are commonly utilized to assess liver function and cellular integrity. Elevated blood ALT and AST levels typically indicate hepatic injury. In our study, we observed a decrease in the levels of both ALT and AST in the liver of R6/2 mice (Figure 1B), and expression levels of several CYPs were also found to be altered (Figure 1C). Several studies have reported that lower ALT and AST levels are associated with dementia [77], increased frailty, and an elevated risk of mortality in the elderly and individuals with illnesses [78,79,80,81]. Lower activity of the ALT and AST enzymes could be associated with liver dysfunction and the short lifespan observed in R6/2 mice [82]. Dysregulation in the transcription of many CYP p450 members is well documented in humans with chronic liver disease along with elevated pro-inflammatory factors [48,49,50]. The findings indicate a decrease in the majority of CYPs with the exception of several increased CYPS in chronic liver diseases. These alterations correlate with the severity of the chronic disease. Furthermore, reduced CYP46 levels in the plasma [83] and putamen of HD patients, as well as in the striatum of R6/2 mice [46], have been previously reported. Our earlier study had also demonstrated a significant lower expression of CYP11A in the testis of BACHD rats [44]. As HTT functions as a transcription factor [84], reduced expression levels of HTT in hepatocytes have resulted in altered liver gene expression patterns [85]. Furthermore, mHTT interacts with several transcription factors and transcriptional activators including wild-type HTT [86]. It raises the question of whether the dysregulated gene expression of CYP members in the liver of R6/2 mice results from directly disrupting the transcriptional process by mHTT. Moreover, the expression level of the transcription factor PPARγ has been found to be markedly reduced in the liver of R6/2 mice, further affecting the transcriptional level of many downstream genes, which are involved in important cell functions [32]. Overall, a lower expression of ALT and AST, as well as transcriptional disturbances in CYP members in the liver of R6/2 mice, demonstrate an impaired hepatic function.

A recent study [87] involving 447,626 participants reported a significant correlation between liver fibrosis and cognitive function as well as the correlation between liver fibrosis and gray matter volume in various brain regions, including the ventral striatum. Notably, serum C-reactive protein (CRP) in the liver, whose level increases in response to inflammation, also correlated with these two neural parameters, indicating an important role of the liver in brain function. The liver communicates with the brain via different mechanisms. Upon liver-to-brain communication pathways, elevated peripheral immune mediators such as TNFα and TGF-β1, whose expression levels are also elevated in R6/2 mice, impact brain neuron activities and trigger behavioral changes. The liver is innervated by vagal afferents that can respond to immune mediators, and this response is further forwarded to the vagal complex in the brain. Moreover, cerebral endothelial cells express various receptors for immune mediators; the binding between circulating immune mediators and receptors initiates their respective signaling pathways. Furthermore, circulating immune mediators and monocytes can directly enter the brain via structures that are highly permeable or structures that lack a blood–brain barrier such as circumventricular organs [88]. Collectively, this highlights the interconnection between the liver and brain.

Consistent with our previous study [37], we observed increased locomotor activities in R6/2 mice in both the dark and light phase. Due to the higher significance and amplitude in the changes, it is likely that ambulatory activities and fine movement were more prominently increased in light phase, and this might imply that this behavioral alteration is associated with sleep disturbances, which was reported previously in R6/2 mice [89,90]. Interestingly, hypolocomotion and sleep disturbances are common features of chronic liver disease [91,92]. Hypolocomotion in rats with chronic liver failure was suggested to be associated with glutamate levels and the activation of glutamate receptors in the substantia nigra [93], while the mechanism underlying sleep disturbances in liver disease patients remains unclear. Thus, we hypothesize that the hepatic pathological changes we observed may have contributed to the neuropathological changes in R6/2 mice. The significant reduction in rotarod performance in R6/2 mice aligns with previous findings [37]. However, in contrast to the highly significant differences in locomotor activities, the changes in rotarod performance between the two treated R6/2 mice groups, reflecting motor coordination and balance, did not reach significance. The disparities in the outcomes of MSC treatment between the two tests could be attributed to the different impact of the liver on various brain regions, each controlling different behavioral tasks. Another possibility might be due to the greater sensitivity of LabMaster, which generated a larger dataset (22 h at each time point), thereby providing increased statistical power. 

Additionally, the treatment with intravenously administered MSCs demonstrated a promising therapeutic effect on the hepatic pathology of R6/2 mice. This was evident by the reduction in pro-inflammatory markers and improved expression levels of liver enzymes that are closely associated with liver function. Fundamentally, the MSC-treated R6/2 mice exhibited enhanced levels of important neuropathological markers, accompanied by improvements in locomotor activities, which are likely associated with ameliorated sleep disturbances (Figure 5 and Figure 6). Remarkably, MSCs were absent in the brain six weeks after the initial administration despite the therapeutic impact on neuropathology and neurobehavior. Instead, a substantial GFP-positive signal (tag of MSCs) was detected in the liver during the postmortem analysis of MSC-injected mice. This observation supports the notion that MSCs have a limited lifespan post-intravenous injection and mainly migrate to the liver and spleen after a short period of time [94,95,96]. Direct homing to the liver maximizes the therapeutic potential of MSCs for addressing liver dysfunction following intravenous injection [35]. Our findings therefore suggest that through the liver–brain axis, improvements in hepatic pathological changes potentially influence neuropathological changes in R6/2 mice. However, it cannot be ruled out that there may be interactions between cerebral endothelial cells and MSCs that remain in the circulation after intravenous administration; this warrants further investigation. Further research on the modulation of liver function, with the exclusion of a direct effect on the brain, needs to be conducted to confirm the importance of hepatic pathology in HD. Furthermore, broadening the cohort to include both genders of mice in our R6/2 HD model would undoubtedly enhance the value of our findings. Additionally, the incorporation of more HD in vivo models, especially across different species, would contribute significantly to confirming the relevance of the hepatic phenomena observed in R6/2 mice to human patients.

MSC therapeutics for neurodegenerative diseases have gained increasing popularity due to the MSCs’ immunosuppressive and anti-inflammatory properties. Furthermore, macrophages play a pivotal role as a key innate component in the immune response against pathogens. Depending on their active state, macrophages can be categorized into type 1 (M1, pro-inflammatory) and type 2 (M2, anti-inflammatory) profiles. Active MSCs play a regulatory role in influencing the phenotypes and activities of both innate and adaptive immune cells. This includes the prevention of B cell proliferation and activation, leading to a reduction in the expression of chemokine receptors and their corresponding ligands [97,98], which were evident in the MSC-treated R6/2 mice (Figure 3). An important mechanism of immune regulation by MSCs involves inducing the transformation of macrophages from an M1 to an M2 state. Our results demonstrate a clear reduction in pro-inflammatory factors indicating the deactivation of M1 macrophages in the liver. However, there are no indications of an increase in M2 macrophages (Figure 3 and Figure 4). The observed stagnation in M2 markers in the MSC-treated mice could be attributed to the overall decrease in macrophage numbers in the liver along with the concurrent decrease in monocytes and dendritic cells (Figure 4), both of which have the potential to differentiate into macrophages. This study sheds light on the intricate immune regulatory effects of MSCs, emphasizing their ability to modulate both innate and adaptive immune responses, which are associated with neurodegenerative diseases. Further research is needed to unravel the precise mechanisms causing these immunomodulatory effects and to explore the full spectrum of MSC-mediated immune regulation in the context of neurodegenerative conditions. 

## 5. Conclusions

Our findings reveal significant hepatic pathology in the R6/2 HD mouse model, which is characterized by mHTT aggregation, inflammation, and alterations in enzymes associated with liver function. The observed hepatic pathological changes are intricately linked to neuropathology, influencing disease progression, and identifying the liver as a potential therapeutic target and diagnostic marker in HD. Supporting this notion, intravenously delivered MSCs exhibited intricate immune regulatory effects in the treated R6/2 mice, coupled with functional improvements in the hepatic cells, as well as neuronal markers and brain function. The evidence presented in this study highlights the interconnectedness of hepatic and neural aspects in HD, suggesting a holistic approach to therapeutic interventions. Further exploration of the immune-modulatory mechanisms and the therapeutic potential of targeting hepatic pathology could offer valuable insights for the development of effective treatments for HD.

## Figures and Tables

**Figure 1 cells-13-00469-f001:**
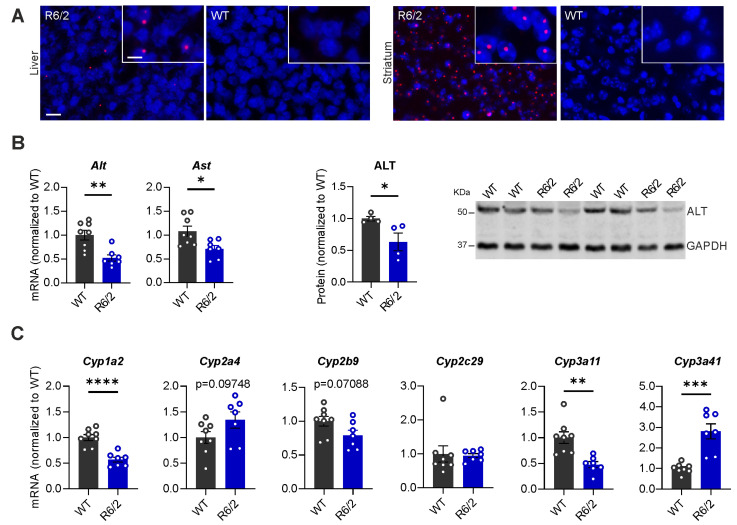
Mutant Huntingtin (mHTT) aggregates develop in the liver, and hepatic pathological changes are induced in R6/2 mice at 10.5 weeks of age. (**A**) mHTT aggregation was detected using fluorescence staining with EM48 antibody (red) in the liver and striatum of R6/2 mice (*n* = 4/group). Nuclei were counterstained with DAPI (blue) (scale bars: inlay, 10 µm; in low-magnification images, 25 µm). (**B**) Quantification of mRNA (Student’s *t*-test, *n* = 8 WT, *n* = 7 R6/2) and protein (Student’s *t*-test, *n* = 4/group) expression levels of ALT and AST in the liver of R6/2 and WT mice. (**C**) Quantification of mRNA expression levels of hepatic cytochrome P450s (*Cyps*) (Student’s *t*-test, *n* = 8 WT, *n* = 7 R6/2). Relative mRNA expression levels were calculated relative to the geometric mean of three reference genes (*Gapdh*, *Sdha* and *Hprt1*), and protein expression levels were normalized to internal control Gapdh. The same methodology was applied for the remaining figures. * *p* ≤ 0.05; ** *p* ≤ 0.01; *** *p* ≤ 0.001; **** *p* ≤ 0.0001.

**Figure 2 cells-13-00469-f002:**
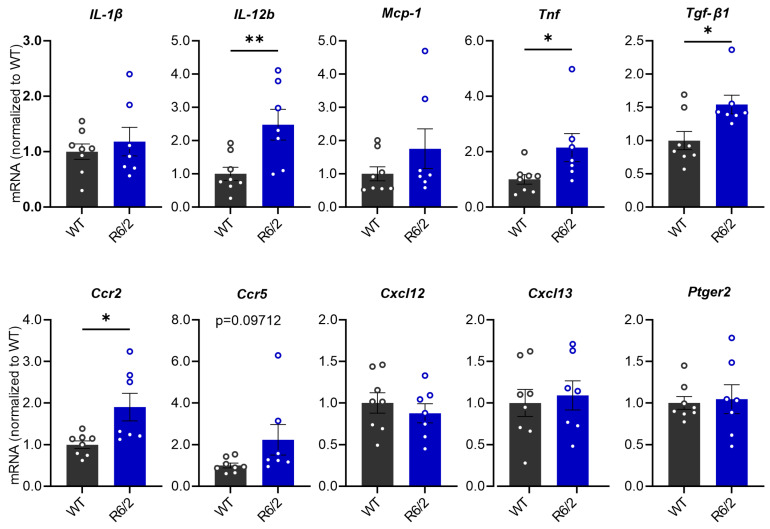
mRNA expression levels of inflammatory regulators in the liver of R6/2 mice compared with WT mice at 10.5 weeks of age. Relative mRNA expression levels were calculated relative to the geometric mean of three reference genes (*Gapdh*, *Sdha* and *Hprt1*), (Student’s *t*-test, *n* = 8 WT, *n* = 7 R6/2). * *p* ≤ 0.05; ** *p* ≤ 0.01.

**Figure 3 cells-13-00469-f003:**
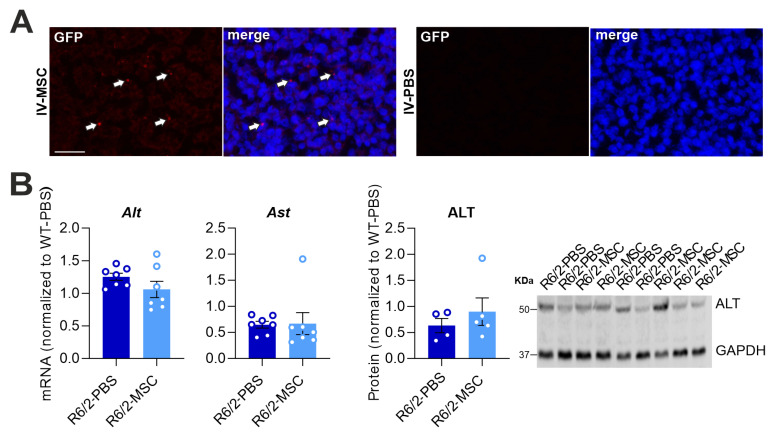
Hepatic pathological changes in R6/2 mice induced by the therapeutic impact of intravenously delivered MSCs. (**A**) Four weeks post-second administration, mice were sacrificed and liver sections were stained with anti-GFP antibody. Abundant GFP signal (red, white arrows) is present in R6/2-MSC mice liver, whereas R6/2-PBS mice do not show any GFP-positive signal (scale bars: inlay, 10 µm). (**B**) Quantification of mRNA expression levels of *Alt* and *Ast* and protein expression of ALT in the liver of R6/2-MSC mice compared with R6/2-PBS control mice (Student’s *t*-test, *n* = 7/group for mRNA analysis, *n* = 4/group for protein analysis). (**C**) Quantification of mRNA expression levels of significantly or slightly altered *Cyps* in R6/2-MSC mice compared with R6/2-PBS control mice (Student’s *t*-test, *n* = 7 R6/2-PBS, *n* = 8 R6/2-MSC). (**D**) Quantification of mRNA expression levels of significant or slightly increased inflammatory regulators in R6/2-MSC mice compared with R6/2-PBS control mice (Student’s *t*-test, *n* = 7/group). Data are normalized to the average of WT sham-treated mice. A value greater than 1 in R6/2-PBS indicates a significant (or trending) increase in R6/2 mice, while a value lower than 1 suggests a significant (or trending) reduction compared to the WT group. Values closer to 1 in the R6/2-MSC group, in comparison to the R6/2-PBS group, indicate a recovery in the MSC-treated group. * *p* ≤ 0.05; ** *p* ≤ 0.01.

**Figure 4 cells-13-00469-f004:**
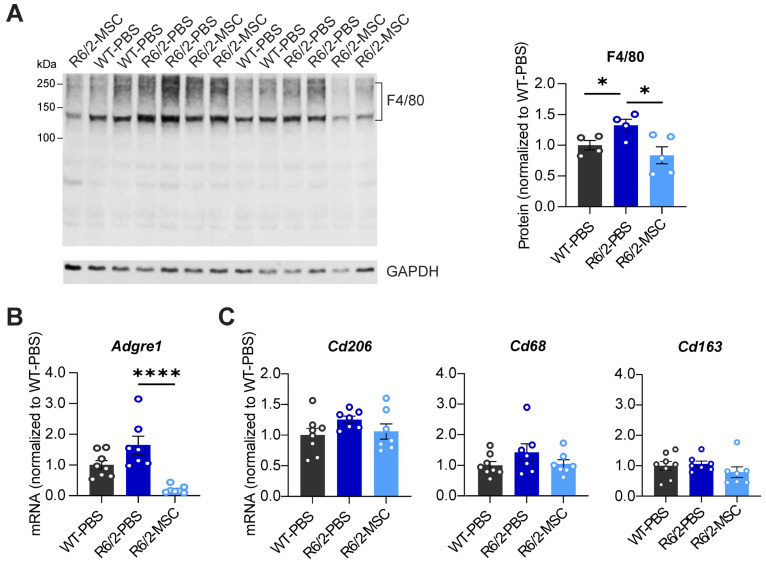
Quantification of macrophage markers in R6/2-PBS control mice compared with WT and MSC-treated mice at 10.5 weeks of age. (**A**) Analyses of F4/80 protein expression in the 3 different treatment groups. (**B**) Quantification of *Adgre1* mRNA expression level. (**C**) Quantification of mRNA levels of M2-type macrophage markers. All statistical tests were performed using Student’s *t*-test. Protein analysis: *n* = 4 WT-PBS, *n* = 4 R6/2-PBS, *n* = 5 R6/2-MSC; mRNA analysis: *n* = 8 WT-PBS, *n* = 7 R6/2-PBS, *n* = 7 R6/2 MSC. Relative mRNA expression levels and protein expression levels were calculated as in Figure 3. * *p* ≤ 0.05; **** *p* ≤ 0.0001.

**Figure 5 cells-13-00469-f005:**
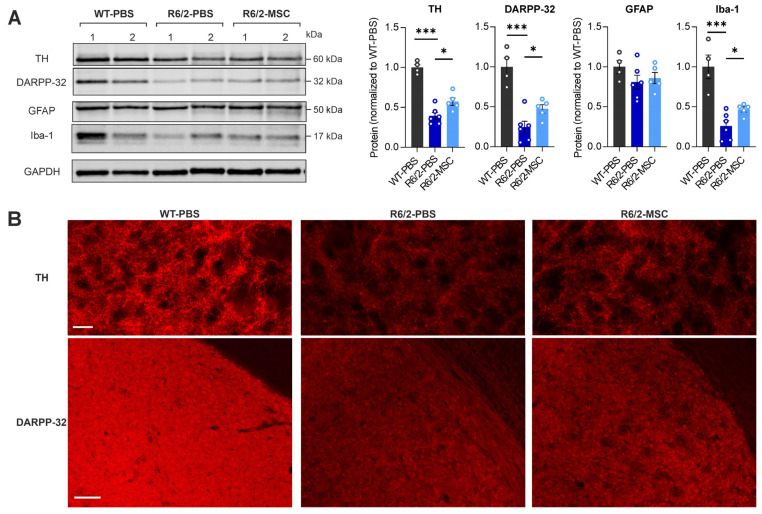
MSC-treatment effect on the brain pathological features of R6/2 mice at 10.5 weeks of age. (**A**) Protein expression levels in the striatum using western blot analyses of the following markers: TH, the marker of striatum-innervated dopaminergic neurons; DARPP-32, the marker of striatal neurons expressing dopamine receptor; GFAP, the marker of astrocytes; Iba-1, a pan-microglial marker. All statistical tests were performed using Student’s *t*-test, *n* = 4 WT-PBS, *n* = 6 R6/2-PBS, *n* = 5 R6/2-MSC. * *p* ≤ 0.05; *** *p* ≤ 0.001. (**B**) Representative images of immunofluorescence staining against respective TH and DARPP-32 confirmed results of western blot analyses presented in (**A**) (*n* = 4/group). Scale bars, 50 µm.

**Figure 6 cells-13-00469-f006:**
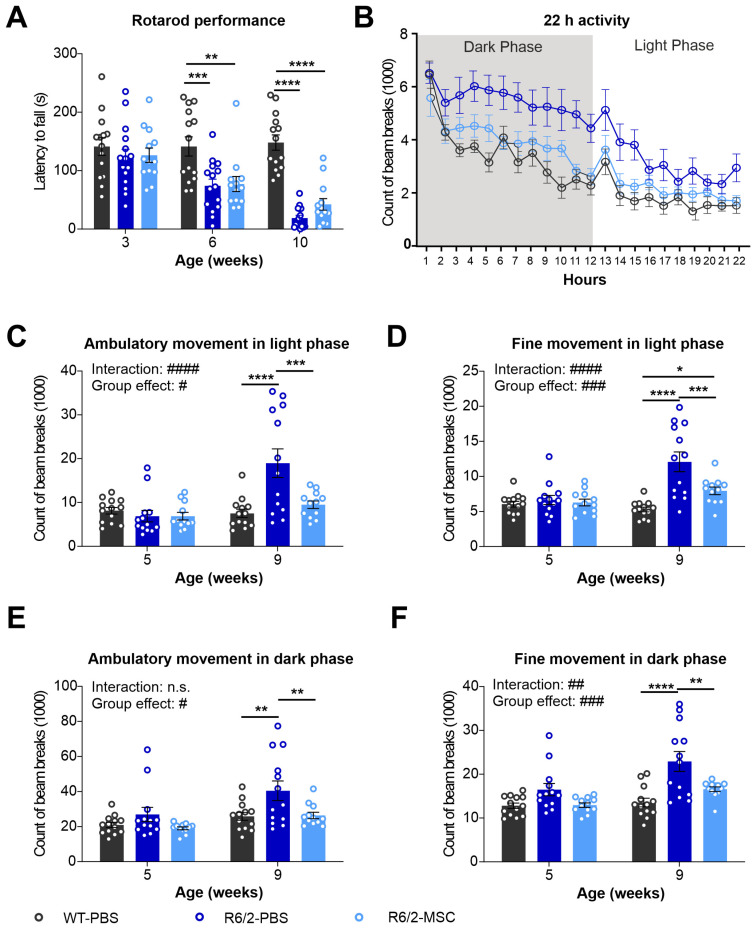
Longitudinal assessment of behavioral phenotypes in R6/2 mice after MSC administration. (**A**) Rotarod test performance of MSC-treated mice at 3 weeks (when the groups were divided), and 6 and 10 weeks of age (time points after MSCs administration) (*n* = 13 for WT-PBS, *n* = 15 R6/2-PBS, *n* = 12 for R6/2-MSC). Locomotor activities of mice were monitored using LabMaster at 5 and 9 weeks of age for 22 h (*n* = 13 for WT-PBS and R6/2-PBS, *n* = 12 for R6/2-MSC). The counts of beam breaks represent the ambulatory activities during the whole recording period (22 h) at 9 weeks of age (**B**), ambulatory activity in the light phase (**C**), fine movement in the light phase (**D**), ambulatory activity in the dark phase (**E**) and fine movement in the dark phase (**F**). All statistical analyses were performed using two-way ANOVA and Tukey’s post hoc test (besides **B**). # indicates the results of two-way-ANOVA analysis; * indicates the results of post-hoc test. n.s., not significant; #/* *p* ≤ 0.05; ##/** *p* ≤ 0.01; ###/*** *p* ≤ 0.001; ####/**** *p* ≤ 0.0001.

## Data Availability

The data presented in this study are available on request from the corresponding author.

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
