# Peer review of "Intravenous MSC-Treatment Improves Impaired Brain Functions in the R6/2 Mouse Model of Huntington’s Disease via Recovered Hepatic Pathological Changes"

_cells, 2024, doi:10.3390/cells13060469_

Round 1
Reviewer 1 Report
Comments and Suggestions for Authors
Libo Yu-Taeger, Ali El-Ayoubi, Pengfei Qi, Lusine Danielyan, Hoa Huu Phuc Nguyen
Intravenous MSC-treatment improves impaired brain functions in the R6/2 mouse model of Huntington's Disease via recovered hepatic pathological changes
COMMENTS FOR THE AUTHOR:
1. The presented research is an original and important for neurology. The manuscript is included all parts which needs for the publication: Abstract, Introduction, Materials and Methods, Results, Discussion, Conclusion, References. The manuscript is included also section Study Limitations.
2. The title clearly and precisely reflects the findings of the manuscript.
3. Abstract is it really a summary, include key findings and have an appropriate length.
4. This study and its introduction reveal the etiology of Huntington's disease. It describes the systematic manifestation of Huntington's disease, including peripheral organ dysfunction and peripheral inflammation. The relevance of the study is undoubtedly due, among other things, to the fairly frequent manifestation of this disease in human populations. The authors also describe in detail the models and methods used to analyze the nature of the disease.
5. The authors described the methods in detail, they are sufficient for reproduction by other researchers. Although some points are not entirely clear to me, because I did not personally conduct them.
6. The general logic of the results is correct, the pictures suggested and located strictly in accordance with the sсheme of section. In my opinion, additional experiments are not required.
7. In the discussion and in the work as a whole, the problem of the use of mesenchymal stem cells in the treatment of various disorders in Huntington's disease is well and clearly characterized. An unexpected result on the therapeutic significance of liver cell repair for the treatment of this disease is very interesting. This result demonstrates the interconnectedness and interdependence of the functioning of various body tissues against each other.
8. The figures correspond to the article’s structure; legends to him explain the drawings. The citation is appropriate, the included the basic publications on the topic.
9. Final comments.
The manuscript is fully consistent to the stated theme. I recommend for publication without any corrections.
Author Response
We express our sincere gratitude to Prof. Gainutdinov for the thoughtful and positive evaluation. We are grateful for his kind words and appreciate the time and effort taken to review our work.
Reviewer 2 Report
Comments and Suggestions for Authors
The article entitled “Intravenous MSC-Treatment Improves Impaired Brain Functions in the R6/2 Mouse Model of Huntington's Disease via Recovered Hepatic Pathological Changes”, is a very interesting and indicative study to find new therapies in Huntington's disease. It is very well constructed, although the biggest problem is that only males are included, this type of bias should not be present in a research. Therefore, a possible publication is proposed after mayor revision. The necessary modifications in each section are detailed below:
Introduction
1. Make a diagram of the relationship between MSC treatment and the relationship with the brain and liver in Huntington's disease.
Material and method
2. Males and females should be included in the article to avoid bias. If you need more time to respond, please let the editors know.
3. Review the entire material and method section carefully, indicating how many animals you used in total and which ones you used for each of the techniques used.
4. Review the entire material and method section carefully and complete the references of the products and devices used and include the reference, commercial company and country.
5. Explain well why you used fresh liver and perfused brain for immunofluorescence. Be careful with the explanation.
6. Indicate the stereotaxic coordinates used in the brain slices.
7. Review all the immunofluorescence photomicrographs and correct the saturation and focus, since most of them are out of focus. It is difficult to ensure in Figure 1 and 2 that the cells are hepatic or striatal. Please post a panoramic photo of the cut to see the correspondence.
8. Likewise, the Western Blot photographs are out of focus. Please check that they are clearly visible when you include them in the article.
9. Enter photos of all raw westerns in supplementary material.
Results
1. The statement you make in point 3.1 “…however, these aggregates were slightly smaller in size compared to the nuclear inclusion bodies in the striatum”, without a comparative study of cell sizes and volumes, cannot be confirmed: You are doing an assumption, that is not a result. Please transform the phrase or omit the comment.
2. Say exactly the location of figure 5B (DARPP-32), put an image at a lower magnification to verify the location.
Discussion and Conclusions
1. The authors should expand on future perspectives and indicate the limitations of the study, addressing the limitations helps provide a balanced perspective and suggests avenues for future research.
Comments on the Quality of English LanguageMinor editing of English language required
Reviewer 3 Report
Comments and Suggestions for Authors
Dear authors, I enjoyed reading the manuscript entitled "Intravenous MSC-Treatment Improves Impaired Brain Functions in the R6/2 Mouse Model of Huntington's Disease via Recovered Hepatic Pathological Changes". This original study, from the preclinical area, carried out on laboratory animals (mice), managed to find correlations between the pathological changes in the liver and those related to the neurological pathology, characteristic of Huntington's Disease. Also, the administration of mesenchymal stromal cells intravenously in R6/2 mice showed an improvement in the status of the animals, which opens new therapeutic perspectives for this condition.
The work complies with the requirements of the journal, the results obtained were highlighted by the large number of clear and valuable figures, offering the reader the possibility to make the reading easier. The impressively large number of bibliographic references used for documentation are in agreement with the chosen topic. Sections: Introduction, Material and methods, Results, Discussion have been described in detail. Although it wasn't mandatory, I appreciate that you also gave us some takeaways.
However, I have a few questions and suggestions:
1. In the Introduction you used the expression "In addition" quite frequently. Where possible, I ask the authors to substitute another expression.
2. Motor function was assessed using the Rotarod test. Why this test? And why only this test? As a rule, a battery of tests is used.
3. To slaughter animals, you used CO2. What was the reasoning in choosing this method of euthanasia?
4. English could be slightly improved.
Comments on the Quality of English Language
English could be slightly improved.
Reviewer 4 Report
Comments and Suggestions for Authors
This was an intriguing and well-written paper. This reviewer feels the majority of the intent was laid out well and that the results section presented findings in an accessible manner. There are a few disappointments encountered with this however along with the conclusion formation in the discussion.
1) The special circumstances of the R6/2 model of HD where as a transgenic, there are multiple instances of mutant huntingtin genes inserted throughout the genome with regular promotion and therefore the animal lacks much ability to control the degree to which the mutant gene is actively expressed by comparison with standard knock-in models where the gene is under standard promotion control and expression of mutated huntingtin protein would be expected to be restricted. It remains unclear if the spread of effects into the R6/2 liver is promoted by this aspect that deviates from that exhibited perhaps by the human condition.
2) The initial findings that seemed intent on exposing the issue represented as a dichotomy in the liver bewteen WT and R6/2 animals presented significant decreases in Alt and Ast mRNA, ALT protein, and Cyp1a2 & Cyp3a11 genes along with an increase in Cyp3a41 gene expression in the R6/2 by comparison to WT. However, the reader is left with little clarity as to what this means and what these genes or proteins do in the liver. But mostly, the disappointment was in the challenge in determining whether the addition of MSCs served to reverse the changes seen (lack of changes across R6/2-PBS versus R6/2-MSC comparisons with Alt, Ast, ALT, and all Cyp genes seemed to elicit little reversal of the phenomena. Tgf-Beta1 and IL-12b did seems to engage what might be considered a reversal of what happened with these inflammatory components in the comparison between R6/2 and WT, but the way these comparisons were juxtaposed was awkward and difficult to flip back and forth to follow. It would be nice if those sorts of events could be brought out more clearly such as how WT was compared with R6/2 with or without MSC additions in figure 4.
3. it was never really made clear why there were two injections made of the MSCs, and how that change may have impacted the behavioral tests performed that seemed to go across the days when the second injection (6 weeks) were made.
4. Regarding behavior results, while granted an interesting finding was determined in the light phase ambulatory movement (also, quite frankly, in the dark phase as well meaning the focus on lack of sleep-related cessation may have been a bit strong related to the presented dichotomy), the rotarod results (or lack of impact of MSCs on this measure) seemed completely missing from the discussion. The light phase focus is perhaps appropriate looking across the whole 22h activity chart, but in fact the impression that MSCs are doing something seemed a bit more prominent in the dark phase to this reviewer. Wouldn't there be a way to discuss how spontaneous movements engaged would be different from coordination of movements (perhaps the latter necessitating some greater neuronal support)?
Round 2
Reviewer 2 Report
Comments and Suggestions for Authors
Thank you very much for the reviews, the work has improved considerably and it would be good to publish with two minor considerations.
Please include the reference of the atlas used in the text and the list of references.
I have not received the graphic summary to which you refer. Possibly it is due to technical issues of the publisher. I ask the editor to send it for verification.
Comments on the Quality of English LanguageMinor editing of English language required
Author Response
Thank you for your comment. We appreciate your input, and we have now included the reference you mentioned, highlighted in green. Additionally, we have attached the graphical abstract for your reference.
